# The Effects of Small-Sided Games and Behavioral Interventions on the Physical and Motivational Outcomes of Youth Soccer Players

**DOI:** 10.3390/ijerph192114141

**Published:** 2022-10-29

**Authors:** Rareș-Mihai Pop, Vlad Teodor Grosu, Emilia Florina Grosu, Alexandru Zadic, Liliana Mâță, Tatiana Dobrescu

**Affiliations:** 1Faculty of Physical Education and Sport, Babeș-Bolyai University, 400347 Cluj-Napoca, Romania; 2Faculty of Mechatronics, Technical University, 400114 Cluj-Napoca, Romania; 3Doctoral School of Physical Education and Sport, Babeș-Bolyai University, 400347 Cluj-Napoca, Romania; 4Faculty of Science, “Vasile Alecsandri” University of Bacău, 600115 Bacău, Romania; 5Faculty of Movement, Sports, and Health Sciences, “Vasile Alecsandri” University of Bacău, 600115 Bacău, Romania

**Keywords:** behavior modification, effort, motivation, SSGs, team sports

## Abstract

The objective of this study was to test the effects of two types of intervention, one based on small-sided games (SSGs), and the other one that had, in addition, a behavioral component consisting of goal setting, public posting, and positive reinforcement. The participants were 16 male soccer players aged 12–14 years old who participated in a couple of training sessions per week between August and November 2021. We used Playr Catapult GPS devices to assess the effect of the intervention on the total distance covered and total sprint distance in the task of playing SSGs. We used Yo-Yo Intermittent Recovery Test Level 1 to test the effect of the intervention on players’ aerobic capacity and the Task and Ego Orientation in Sport Questionnaire to investigate levels of motivational climate. Multiple two-way mixed ANOVAs were conducted and the results indicated that our intervention had a positive effect on the total distance and total sprint distance covered during SSGs. For aerobic capacity, even if the results were not significant, they indicate a high effect size. The effect of the intervention on task-oriented motivational climate and ego-oriented motivational climate was not significant. Discussions focus on the application of the intervention in team training settings.

## 1. Introduction

Soccer is a sports game that involves variable effort intensity, characterized both by high-intensity specific motor actions such as sprints or jumping, and also by motor actions of medium or low intensity such as light running or walking, thus involving both systems responsible for the exercise capacity, the aerobic and anaerobic system [1,2]. Even though elite soccer players average an effort intensity close to the anaerobic maximum threshold, reaching a maximum heart rate (Hf_max_) of 80–90% and a maximum oxygen absorption (VO_2max_) of 70–80%, most of the energy used during matches and training is produced by the aerobic system [3]. Therefore, we can say that players who practice the game of soccer need a high aerobic and anaerobic effort capacity to be able to cope with the demands of the modern game. Moreover, using the ball when training for aerobic capacity has additional benefits related to improving players’ technique, which is especially important in the developmental process of youth; however, this desideratum is rarely fulfilled in sports training [4].

According to Snow [5], youth players have an intrinsic desire to play the game, and activities should be game-specific because they are fun and age-appropriate. Small-sided games (SSGs) are often used to improve the aerobic capacity of soccer players. They are “an adjusted form of official games that are often used in training scenarios to introduce a specific tactical issue to team sports players” [6]. To be effective, several variables related to these games must be handled properly. However, it appears that using the ball to increase the capacity for effort does not yield results only in improving the specific technique of the game in youth soccer players. Experimental studies show that athletes could also enjoy benefits related to the level of pleasure experienced in sports activity [7]. Therefore, it is possible and, in fact, necessary, to reduce the share of exercises without a ball for the development of effort capacity and to replace them with exercises that use the soccer ball. Nevertheless, the results of studies that investigated the use of soccer games and ball drills to increase the aerobic capacity of soccer players are slightly contradictory. On the one hand, there have been several studies that have shown that 11-a-side soccer games used as a means to develop exercise capacity in training did not have a high enough intensity to improve those indices [2,8]. On the other hand, although the soccer game played under normal conditions may not be intense enough to develop aerobic capacity, it seems that SSGs have a high enough intensity to develop aerobic capacity and thus lend themselves well as an alternative to training that uses running without the ball [4,9]. In order to increase the intensity of SSGs, the authors used a wide range of strategies [10]. Among these strategies, the most used were altering the number of players in a team and altering the size of the pitch [11]. In terms of team size, SSGs with fewer players seem to have a stronger effect on aerobic capacity, while the larger the size of the pitch in relation to the number of players, the more pronounced the effect of training on aerobic capacity [10]. According to Sarmento et al. [10], other variables associated with SSG intensity were: the use of different goal posts, conditioning the ball touches, changing the training regimen, use of coach encouragement, changing the type of marking, using floaters, and other artificial rules.

In studies that targeted psychological and behavioral outcomes, but also in practice and in studies related to sports performance, authors and practitioners used behavioral modification techniques to change the frequency or intensity of certain behaviors [12]. Even though behavioral modification techniques have high applicability in the field of sports performance, as proven by the numerous studies carried out on samples from different sports [13], none of those studies have targeted the intensity of effort during SSGs. Recent studies are based on measuring the effects of small-sided games on different components: technical skills [14,15], VO_2max_ and skills improvement [16], heart rate, rating of perceived exertion, and running demands in professional soccer players [17], the physical fitness [18,19,20,21,22], the development of endurance [23], hamstring eccentric strength [24].

An example of a behavior modification technique is monitoring. Monitoring not only has the function of providing information about the investigated behavior but can also represent an actual intervention due to the appearance of reactivity [12]. For example, one study showed that monitoring the daily level of physical activity by sedentary adults led to a significant increase in physical activity levels [25]. Goal setting refers to the process of choosing relevant objectives, aims, or personal standards that individuals want to achieve [26]. In a review of the literature that investigated the use of goal setting in the sports field, results showed that to achieve superior performances, the most effective are specific goals set both for short- and long-term and of moderate to high difficulty [27]. As for the choice of athletes, it seems that they want to have moderately difficult goals [28].

Public posting is a feedback procedure in which individual performance is visible to others, and which promotes social comparison [29]. Even if goal setting and public posting were used in multiple interventions to improve athletic performance, most target behaviors in those interventions were related to technical performance [13]. Positive reinforcement refers to the application of an immediate consequence with positive valence after a desirable behavior to strengthen that behavior [12]. According to Weinberg and Gould [30], reinforcements need to be valued by the athletes, and coaches might use a wide range of rewards such as praise, a smile, a pat on the back, trophies, medals, or different activities. Another behavioral modification technique that works on the principle of positive reinforcement is the token economy. In the token economy, a person receives a token following a desirable behavior, which can be exchanged later for backup reinforcers [12]. Token economy has also been used before in a few studies in exercise settings, mostly in the context of physical education [31]. This study aimed to test the effect of an intervention composed of monitoring, goal setting, public posting, and positive reinforcement on total distance covered, total sprint distance, aerobic capacity, and task- and ego-oriented motivational climate.

The first specific goal was to test the intervention’s effect on total distance and total sprint distance covered during the task of SSGs. The second specific objective of our study was to test the effectiveness of the intervention on the aerobic capacity of athletes. The third specific objective was to check whether the proposed intervention will have an effect on increasing the level of the task- and ego-motivational climate perceived by the athlete and within the team. According to the achievement goal theory [32], athletes define their success according to the behavior of the coach, which can encourage two types of motivational climate: task-oriented or ego-oriented. In the case of a task-oriented motivational climate, coaches encourage the progress and effort made by athletes, and in the case of an ego-oriented motivational climate, coaches encourage obtaining victory and overcoming opponents and colleagues. The task-oriented motivational climate is associated with numerous benefits among athletes, such as high levels of pleasure and positive emotions, high levels of performance and self-reported progress, and positive relationships with colleagues (for a review see [33]).

## 2. Materials and Methods

In our study, we used a mixed two-factor experimental design. This design tests the differences between two or more independent groups, while the participants of the groups are also subjected to repeated measurements [34]. The first independent variable was a within-subjects factor represented by repeated measurements (before and after the intervention). The second independent variable was a between-subjects factor naming the type of intervention, which had two groups. The first experimental group participated in an intervention based on SSGs, and the second experimental group participated in an intervention that had in its composition, besides SSGs, a behavioral modification component. Our study had five dependent variables: total distance covered, total sprint distance, aerobic capacity, and task- and ego-oriented motivational climate.

### 2.1. Participants

The participants were 16 male soccer players from Cluj-Napoca, Romania, aged 12–14 years old (M = 13.24, SD = 0.68), with an average height of 165.5 cm (SD = 9.75), an average weight of 51.75 kg (SD = 10.65), and an average BMI of 19.43 (SD = 2.59). They participated in a couple of training sessions per week between 17 August 2021, and 9 November 2021. The participants were selected based on their availability and willingness to participate, resulting in a convenience sample, which is a commonly used method, despite the limits related to external validity [34].

First, the team coach was contacted to explain the purpose of the study and to obtain their consent. In the case of the refusal of one of the coaches to participate in the study, another team was selected. After obtaining the coach’s consent, we contacted the parents of each athlete to obtain their informed consent and to inform them about the procedure and the fact that participation was voluntary, and the athletes could withdraw at any time from participating in the study. Only those athletes who agreed to participate and whose parents expressed their consent to participate in writing took part in the study. The procedure is similar to that used in other studies that have investigated groups of athletes under the age of 18 [35]. Due to the small number of participants, to obtain two homogeneous groups, the players were distributed in one of the two experimental groups using matched pairs, based on the aerobic capacity initial testing results [36].

### 2.2. Research Instruments

We used Playr Catapult GPS devices to assess the effect of the intervention on the total distance covered and total sprint distance. The literature suggests that GPS devices should work at a minimum 10 Hz sampling frequency in order to achieve reliable and valid results; however higher sampling frequency does not provide further benefits [37]. According to the manufacturer, this model function at a frequency of 10 Hz and is able to record up to 1250 movements/second [38].

Yo-Yo Intermittent Recovery Test Level 1 (YYIRT1) was used to measure estimates of players’ aerobic capacity. This test is considered a valid and specific measure of soccer players’ aerobic capacity [39]. According to Bangsbo et al. [40], there is a significant correlation between YYIRT1 and VO_2max_ (r = 0.70; *p* < 0.05).

Levels of the task- and ego-motivational climate were assessed with the Task and Ego Orientation in Sport Questionnaire (TEOSQ) [41]. This instrument has acceptable internal consistency in sports settings α = 0.79 for task orientation, α = 0.81 for ego orientation [42].

### 2.3. Procedure

After the athletes and their parents signed the informed consent forms, the first step was to collect the athletes’ anthropometric measurements. After that, the athletes filled in questionnaires related to the motivational climate and performed the Yo-Yo Intermittent Recovery Test Level 1 (YYIRT1) to measure their aerobic capacity. In the following training sessions, they participated in a task of four series of 4 vs. 4 SSGs with a duration of 5 min and breaks of 2.5 min, and we used PLAYR Catapult GPS devices to test their total distance covered and total sprint distance [38]. Finally, based on results obtained from the aerobic capacity test, we distributed the participants into the two experimental groups using matched pairs to obtain two homogenous groups.

#### 2.3.1. Experimental Group 1 (SSGs)

The experimental task consisted of SSGs, and the size of the team was set with either 2 or 4 players. In a meta-analysis that examined the use of SSGs in soccer, the results showed that SSGs with a smaller number of players lead to higher heart rate levels than those with a relatively high number of players [11]. The intervention took place for 12 weeks (between 17 August 2021, and 9 November 2021) and started in pre-season, which is an optimal period for the development of players’ aerobic capacity [43]. The duration of the intervention was similar to that used in other studies that aimed to increase soccer players’ aerobic capacity and to studies that used behavioral interventions in motor tasks [25,44].

The weekly frequency of training sessions was two sessions per week, with a total of 22 training sessions, out of which 6 training sessions were used mainly for testing purposes and 16 for intervention. According to Baquet et al. [44], 2 training sessions per week with a duration between 30 and 60 min might be enough to improve aerobic capacity. As regards the duration of the intervention, it appears that it does not play a decisive role, as similar results have been achieved in interventions lasting 4 weeks compared to interventions lasting 18 months [44].

The games were played at small goals, with no goalkeepers. No other task constraints or rule modifications were used. The coach explained to the players that the aim of the games was to develop effort capacity and emphasized the importance of effort exertion during the games.Phase 1—increasing the volume of aerobic capacity training

According to Bompa and Haff [45], as a general rule, to improve aerobic capacity, coaches need to first increase the volume of training, and then decrease the volume and raise the intensity of training. In this phase, which lasted for seven training sessions, we used 4 vs. 4 SSGs with a duration of 5 min and breaks of 2.5 min and the size of the pitch was 40 × 22 m (110 sqm/player). During this period of the intervention, we focused on the gradual increase in volume. Therefore, in the first training session, participants played 4 series of SSGs, in the following four training sessions they played 4 to 6 series, and in the last two sessions, they played 6 series.Phase 2—increasing the intensity and decreasing the volume of aerobic capacity training

In this phase, which lasted for nine training sessions, we focused on increasing the intensity of the SSGs. For this reason, we used six series of 2 vs. 2 SSGs with a duration of 3 min at each training. The 2 vs. 2 SSGs promote higher levels of training intensity than the 4 vs. 4 SSGs [46]. Moreover, in this phase, we gradually increased the size of the pitch from 28 × 16 m (112 sqm/player) in the first five training sessions to 40 × 16 m (160 sqm/player) in the last four training sessions. According to Sarmento et al. [9], SSGs played on pitches with larger areas relative to the number of players, lead to higher physiological loads and training intensity.

Manipulation of the work-to-rest ratio is another variable that can influence physiological responses during SSGs [47]. As an additional strategy to further increase the intensity, we progressively changed the work-to-rest ratio thusly: in the first five training sessions the duration of breaks was 1.5 min, and in the last four training sessions the duration of breaks was 1 min. The whole training protocol was designed taking into account recommendations from the literature about the use of SSGs for increasing the aerobic capacity of soccer players [43,46,48]. Table 1 shows a brief summary of the variables used.

#### 2.3.2. Experimental Group 2 (SSGs + Behavioral Intervention)

Athletes from this experimental group followed the same training protocol described above. However, in addition to the training regimen of the athletes from the previous group, they received a behavioral intervention composed of behavior monitoring, goal setting, public posting, and positive reinforcement. For behavior monitoring, we used Playr Catapult GPS devices and we tracked two variables: total distance covered and total sprint distance [39]. The results of the behavior monitoring for every training session were displayed in a visible spot near the reserve bench, so every team member was aware of the results of players from this group. This strategy is known as public posting, and it was used before in other studies related to sports performance [49].

We also used individual- and group-level goals related to the total distance covered and total sprint distance. Initial goals for each athlete were set based on the performance recorded in both aerobic capacity testing and initial SSGs testing. Goals were adjusted from time to time during the intervention according to training volume, type of SSGs, or in case the researcher observed that they were inadequate for a specific athlete. Each time an athlete achieved a goal, that goal progressively increased in difficulty. Initial goals for each athlete were set at a level of 10% higher than the performance recorded in the initial SSGs testing. In situations where we observed athletes who had low performance on baseline SSGs testing but a high performance on baseline aerobic capacity testing, the baseline goal was further increased. Each time an athlete achieved a goal, that goal progressively increased in difficulty. For total distance covered the goals increased by 100 m each time they were met, and for sprint distance, they increased by 10 m on every goal achievement. Goals were adjusted from time to time during the intervention according to training volume, type of SSGs, or in case the researcher observed that they were inadequate for a specific athlete. Additionally, for group-level goals, every time at least 10 goals out of the total of 16 set for the team were met, the points from that training session were doubled for every athlete. Moreover, in order to reward the effort and progress made by athletes, we used positive reinforcement. One of the reinforcements that we used was praise. Thus, every time the coach noticed an athlete showed a high and increased level of effort, e.g., a sprint or a high-intensity duel, he praised the athlete. Another reinforcement was represented by the awarding of the captain’s armband during training to the athlete who made the greatest progress in total distance covered and total sprint distance in the previous training session. Praise is a generalized conditional reinforcement because it is associated with the appearance of other reinforcements in various situations [12]. We believe that the captain’s armband is also a generalized conditional reinforcement for most soccer players because it is associated with other reinforcements that the athlete receives such as the respect of coaches and teammates.

In order to better take into account, the athletes’ preferences in reinforcements received, the third type of reinforcement was provided with the use of token economy. Thus, as described above, for each goal fulfilled by an athlete, he received a token, and if participants fulfilled the team-level goal as well, the tokens for that training session were doubled. According to the rules of the token economy, at any time, athletes could exchange the points received with their favorite products or activities, depending on their value. The backup reinforcers from which athletes could choose were: a pair of soccer socks (14 points), a local team soccer cap (16 points), two local team game tickets (18 points), a local team soccer scarf (18 points), a training shirt (19 points), a pair of soccer shin pads (21 points), an agility ladder (22 points), buy an orange juice for every player (36 points), a local team soccer short (36 points), a soccer ball replica (43 points), a local team match shirt (50 points), or a pair of soccer boots (50 points).

### 2.4. Data Analysis

Multiple two-factor mixed ANOVAs were conducted to test the effect of the behavioral intervention on each dependent variable: total distance covered, total sprint distance, aerobic capacity, and task- and ego-oriented motivational climate. We preferred to use multiple ANOVAs on each dependent variable instead of MANOVA because our research questions were univariate. According to Huang [50], when this is the case, there are no further benefits in using MANOVA or other multivariate tests. Cohen [51] recommends using ANOVA in the case of small samples because it is more powerful in detecting differences between means.

## 3. Results

Table 2 shows the participants’ descriptive statistics such as the mean, the standard deviation, and the number of participants for each condition of the dependent variables.

The results of two-way mixed ANOVA (Table 3) revealed a significant difference for the interaction between “type of intervention” and “repeated measures” on the dependent variable “total distance covered” [F(1,14) = 6.235; *p* = 0.026; ηp^2^ = 0.308]. This means that in regard to this variable, participants who received a combined intervention (SSGs and behavioral intervention) improved their results further than the results of the SSGs group alone by the end of the intervention.

Similarly, looking at Table 4, we can see that for the dependent variable “total sprint distance” the interaction effect was also statistically significant [F(1,14) = 4.781; *p* = 0.046; ηp^2^ = 0.255].

Regarding the dependent variable “aerobic capacity”, even if the interaction effect was not significant [F(1,14) = 2.861; *p* = 0.113], results reveal a large effect size (ηp^2^ = 0.170) (Table 5). Furthermore, upon analyzing the main effects, we found a statistically significant difference between the average score for the aerobic capacity test at the beginning of the intervention compared to the end of the intervention, suggesting that the main effect of the intervention was significant [F(1,14) = 47.002; *p* < 0.001; ηp^2^ = 0.770].

In contrast, as shown in Table 6, independent variables produced no significant interaction on task-oriented motivational climate [F(1,14) = 0.014; *p* = 0.908; ηp^2^ = 0.001]. Regarding the main effects, neither the “task-oriented motivational climate” [F(1,14) = 1.110; *p* = 0.310; ηp^2^ = 0.073], nor the “type of intervention” were significant [F(1,14) = 0.021; *p* = 0.886; ηp^2^ = 0.002].

Finally, Table 7 illustrates that both the interaction [F(1,14) = 0.175; *p* = 0.682]; ηp^2^ = 0.012 and main effects on ego-oriented motivational climate were not significant.

## 4. Discussion

In our study, we aimed to evaluate the effect of two types of interventions on variables related to athletes’ effort, effort capacity, and motivation. The first intervention was based on SSGs, and the second had, in addition to SSGs, a behavioral intervention component. In terms of the physical variables, i.e., total distance covered, and total sprint distance, both experimental groups improved these indicators at the end of the intervention compared to the beginning of the intervention. Furthermore, in congruence with our hypothesis, the second experimental group, which received a behavioral intervention in addition to SSGs, improved its performance on these parameters more compared to the first experimental group. The fact that both the total distance covered and the total sprint distance were improved during the SSGs at the end of the intervention looks promising in terms of the ability to increase players’ effort during these games. This is because, in practice, there is usually an inversely proportional relationship between total distance covered and total sprint distance based on the relationship between volume and intensity in sports training [45].

Regarding the estimated aerobic capacity, results show that both experimental groups improved their performance at the end of the intervention. On the other hand, we expected the behavioral intervention group to improve their aerobic capacity further compared to the other group at the end of the intervention compared to pre-intervention levels. However, that was not the case since no significant differences were registered between the two groups or for the interaction. However, even though the differences in the interaction effect were not statistically significant, in terms of effect size, they were meaningful. The reasons for these results could be explained by the small sample size. Another explanation could be related to the fact that we used a field test to determine an estimated level of aerobic capacity, which is subject to error. Future studies should use objective methods of determining aerobic capacity such as ergo spirometry [52]. Finally, in this study, the training sessions for the development of aerobic capacity had a weekly frequency of 2 sessions per week. In their other 2 weekly training sessions, as well as in the friendly games they played, the participants from both experimental groups were assigned the same training protocol, and hence, had the same effort expenditure. We believe that this could have indirectly contributed to the development of aerobic capacity for all players, making differences between groups more difficult to notice.

Our behavioral intervention proposed to emphasize the athlete’s effort expenditure and the progress achieved in their aerobic capacity. Therefore, we expected athletes from the combined intervention group to have a higher task-oriented and a lower ego-oriented motivational climate level than the athletes from the SSGs group at the end of the intervention. Yet, that was not the case, since no significant differences were recorded either for the interaction, between the groups, or between the two repeated measurements. An explanation for this lack of meaningful results in the case of motivational climates could be represented by the use of self-reported data. Future studies ought to assess motivation from different perspectives and not be limited to the use of questionnaires.

One of the limits of our study was the use of a sample of convenience. Taking into consideration that the sample was made up of male soccer players aged between 12 and 14 years old from urban areas, the results obtained must be extrapolated with caution to other subjects. In order to increase the external validity and to generalize the results to other categories of the population, it is necessary to replicate the study and also include subjects of the female gender, of different ages, from different backgrounds, and with different levels of experience in practicing the game of soccer.

As mentioned above, another limitation of this study was represented by the small sample size. This was mainly caused by the limited number of available GPS devices and by the need to use them in each training session and could have led to threats to internal validity. Brysbaert [53] recommends at least 27 participants in each group to achieve high power. Future studies should look to gather data from a large sample size chosen randomly. The lack of a control group was another limitation of our study. We used as a comparison group an intervention based on SSGs, which is also effective in improving aerobic capacity, according to the literature [10]. A true control group could not be used for reasons of ethics and feasibility. In other words, it would not have been appropriate for a group of athletes not to develop their aerobic capacity during the pre-season to serve as a control group.

We believe that a strong point of this intervention is represented by its possible implications on the theory of sports training. If athletes would exert a sufficiently high effort during SSGs, the development of aerobic capacity could be done through such games, giving up the runs used in the classical training methods and hence using the ball more. Consequently, using the ball more often during training could have an impact on the level of pleasure experienced by athletes and on improving their playing technique [4,7]. On the other hand, the use of the proposed program in the case of children training at present could be problematic due to the high costs it entails and the time and resources required to implement the program.

## 5. Conclusions

This study ought to assess the effects of two types of SSGs interventions on multiple physical and motivational outcomes of youth soccer players. For physical outcomes, the results suggest that adding a behavioral component to established strategies can bring additional benefits to training intensity. We believe that this could contribute to aerobic capacity training using SSGs. However, in the case of motivation, the results were inconclusive. The use of wearable sensors in soccer training has gained a lot of popularity recently and provides new opportunities for practitioners who try to improve players’ performance. Overall, we consider that our results have practical value and that the coaches could integrate the strategies used in our research: goal setting, public posting, and token economy, along with other validated strategies, to increase the intensity of SSGs. 

Future studies should address multivariate research questions and further investigate the nature of the relationship between motivation, effort, and effort capacity in the context of SSGs using different study designs, more participants, and objective measures.

## Figures and Tables

**Table 1 ijerph-19-14141-t001:** Summary of variables used to adjust the volume and intensity of SSGs.

Phase	SSGs	Pitch Size	Duration of Series	Rest	Number of Series
Phase 1	4 vs. 4	40 × 22 m	5 min	2.5 min	4–6 series (gradually increasing)
Phase 2	2 vs. 2	28 × 16 m and 40 × 16 m	3 min	1.5 min to 1 min	6 series

**Table 2 ijerph-19-14141-t002:** Descriptive statistics of study participants.

Dependent Variable	Intervention	Repeated Measures	Mean	Variation Change	Std. Deviation	*n*
Total distance covered	SSGs	Before	2058.75		137.782	8
		After	2193.63	+6.55%	197.678	8
	SSGs + behavioral	Before	2091.25		156.519	8
		After	2507.50	+19.90%	158.001	8
Total sprint distance	SSGs	Before	27.00		22.155	8
		After	56.00	+107.40%	38.307	8
	SSGs + behavioral	Before	25.63		20.667	8
		After	100.63	+292.62%	56.830	8
Aerobic capacity	SSGs	Before	43.287		2.381	8
		After	46.150	+6.61%	2.945	8
	SSGs + behavioral	Before	43.713		2.733	8
		After	48.450	+10.83%	3.295	8
Task-oriented climate	SSGs	Before	4.357		0.425	8
		After	4.446	+2.04%	0.428	8
	SSGs + behavioral	Before	4.339		0.381	8
		After	4.410	+1.63%	0.345	8
Ego-oriented climate	SSGs	Before	2.583		1.003	8
		After	2.729	5.65%	0.695	8
	SSGs + behavioral	Before	3.041		1.125	8
		After	2.958	−2.72%	0.640	8

**Table 3 ijerph-19-14141-t003:** Analysis of variance for dependent variable “total distance covered”.

Source of Variation	Degrees of Freedom	Sum of Squares	F	*p*-Value	ηp^2^
Total distance	1	607,477.531	23.921	0.000	0.631
Type of intervention	1	239,951.281	8.459	0.011	0.377
Total distance × Type of intervention	1	158,343.781	6.235	0.026	0.308

**Table 4 ijerph-19-14141-t004:** Analysis of variance for dependent variable “total sprint distance”.

Source of Variation	Degrees of Freedom	Sum of Squares	F	*p*-Value	ηp^2^
Sprint distance	1	21,632.000	24.437	0.000	0.636
Type of intervention	1	3741.125	1.946	0.185	0.122
Sprint distance × Type of intervention	1	4232.000	4.781	0.046	0.255

**Table 5 ijerph-19-14141-t005:** Analysis of variance for dependent variable “aerobic capacity”.

Degrees of Freedom	Sum of Squares	F	*p*-Value	ηp^2^
1	115.520	47.002	0.000	0.770
1	14.851	1.070	0.318	0.071
1	7.031	2.861	0.113	0.170

**Table 6 ijerph-19-14141-t006:** Analysis of variance for dependent variable “task-oriented motivational climate”.

Source of Variation	Degrees of Freedom	Sum of Squares	F	*p*-Value	ηp^2^
Task-oriented	1	0.052	1.110	0.310	0.073
Type of intervention	1	0.006	0.021	0.886	0.002
Task-oriented × Type of intervention	1	0.001	0.014	0.908	0.001

**Table 7 ijerph-19-14141-t007:** Analysis of variance for dependent variable “ego-oriented motivational climate”.

Source of Variation	Degrees of Freedom	Sum of Squares	F	*p*-Value	ηp^2^
Ego-oriented	1	0.008	0.013	0.911	0.001
Type of intervention	1	0.945	0.960	0.344	0.064
Ego-oriented × Type of intervention	1	0.105	0.175	0.682	0.012

## Data Availability

Not applicable.

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
