# Peer review of "The Effects of Small-Sided Games and Behavioral Interventions on the Physical and Motivational Outcomes of Youth Soccer Players"

_ijerph, 2022, doi:10.3390/ijerph192114141_

Round 1

Reviewer 1 Report

Most of the twenty or so bibliographic references prior to 2009, which were used, namely to characterize the prevalence of some variables in Football, must be updated or properly framed in time.

It is advisable to use more studies on SSGs, namely those that have been carried out in recent years.

The following procedures (lines 231 to 235) should be further specified:

“Initial goals for each athlete were set based on the performance recorded in both aerobic capacity testing and initial SSGs testing. Goals were adjusted from time to time during the intervention according to training volume, type of SSGs, or in case the researcher observed that they were inadequate for a specific athlete. Each time an athlete achieved a goal, that goal progressively increased in difficulty.”

Author Response

Comment:

Most of the twenty or so bibliographic references prior to 2009, which were used, namely to characterize the prevalence of some variables in Football, must be updated or properly framed in time.

It is advisable to use more studies on SSGs, namely those that have been carried out in recent years.

The following procedures (lines 231 to 235) should be further specified.

Reply:

We changed part of the bibliographic references and we updated some of them. We replaced the mentioned paragraph with the following one:

”Initial goals for each athlete were set at a level of 10% higher than the performance recorded in the initial SSGs testing. In situations where we observed athletes who had a low performance on baseline SSG testing but a high performance on baseline aerobic capacity testing, the baseline goal was further increased. Each time an athlete achieved a goal, that goal progressively increased in difficulty. For total distance covered the goals increased by 100 meters each time they were met, and for sprint distance they increased by 10 meters on every goal achievement. Goals were adjusted from time to time during the intervention according to training volume, type of SSGs, or in case the researcher observed that they were inadequate for a specific athlete.”

Reviewer 2 Report

General comments

This article explores the effects of two SSG interventions on the players’ total distance covered, sprinting distance, aerobic capacity (YOYOIRTL1) and a Task and Ego Questionnaire. The results suggested a higher benefit towards the group that performed the SSG combined with behavioural components (goal setting, public posting, and positive reinforcement). The study is interesting, mainly as it provides practical information regarding strategies to foster the players’ physical development. Therefore, the authors should be congratulated for developing such piece of work. Despite that, I have some suggestions that should be addressed. 

Specific comments

Abstract - Keywords

Please consider changing the keywords to avoid repeating the same words presented in the title. For example, soccer may be changed by team sports. 

Introduction

L47-51. Please consider adjusting this paragraph, as it is, it seems to indicate that authors will explore SSG under the technical domain. May rather consider suggesting that, in addition to the physical skills, small-sided games also develop technical actions, which are key during their developmental process.

L63-64. I would recommend the authors explore and include studies related to youth. For example, a common rule used is limiting the number of touches during SSG, which seems to foster the players' technical abilities. However, when they do not possess the perceptual and motor skills to do so, the game intensity decreases. 

L. 70. Avoid using “among” twice. 

L.77. Please consider changing “modify” change.

L.98. Add a full stop after the reference. 

While the introduction is well-written with relevant literature, I am missing more information regarding youth players. I mean, the introduction is solid but apart from that SSG helps youth players to develop technical skills, it is not clear why will the authors study this age group. 

Methods. 

L140. Please provide additional information regarding the participants. What was their height, weight, and playing experience? 

L.154. What initial tests? Is there any difference between the players’ profiles between groups?

L. 156. Please provide a reference for the validity of this system. 

L180-182. Could it be expected that players improved their performance, even without the intervention, as it is likely that being exposed to training routines (as those since the start of preseason) would contribute to physical fitness enhancement?

L.190. Please consider including a table with the games performed, or a figure describing the protocol procedures. 

L. 195-202. Which type of rules were used in the SSG? Was there any specific information from the coach? Did it contain goalkeeper? Further information on how these SSG were designed is required. 

L230-246. My main concern here, although I do understand its relation to the authors' study aim, is that running more it is not always running better from the SSG perspective. For reference, please see: 10.1080/15438627.2017.1393754

L.240. What is considered a high level of effort? What was the level of the coaches? Were they able to perceive the appropriate time to encourage the players? How was this controlled by the research team?

Results. 

Please consider including the percentage variation change between each pre-post measurement. 

Discussion

L-328-329. Did the authors control the time performed on friendly and competitive matches by the players from both groups? Were there any differences? Or even maturational differences?

Author Response

General comments

This article explores the effects of two SSG interventions on the players’ total distance covered, sprinting distance, aerobic capacity (YOYOIRTL1) and a Task and Ego Questionnaire. The results suggested a higher benefit towards the group that performed the SSG combined with behavioural components (goal setting, public posting, and positive reinforcement). The study is interesting, mainly as it provides practical information regarding strategies to foster the players’ physical development. Therefore, the authors should be congratulated for developing such piece of work. Despite that, I have some suggestions that should be addressed. 

Specific comments

Abstract - Keywords

Please consider changing the keywords to avoid repeating the same words presented in the title. For example, soccer may be changed by team sports. 

Reply:

We replaced ”small-sided games” with ”SSGs” and ”soccer” with ”team sports”, as suggested.

Introduction

L47-51. Please consider adjusting this paragraph, as it is, it seems to indicate that authors will explore SSG under the technical domain. May rather consider suggesting that, in addition to the physical skills, small-sided games also develop technical actions, which are key during their developmental process.

Reply:

We removed the mentioned paragraph and replaced it with the following one:

”Moreover, using the ball when training for aerobic capacity has additional benefits related to improving players technique, which is especially important in the developmental process of youth; however, this desideratum is rarely fulfilled in sports training [4].”

L63-64. I would recommend the authors explore and include studies related to youth. For example, a common rule used is limiting the number of touches during SSG, which seems to foster the players' technical abilities. However, when they do not possess the perceptual and motor skills to do so, the game intensity decreases. 

Reply:

We added a new sentence: According to Sarmento et al. [9], other variables associated with SSGs intensity were: the use of different goal posts, conditioning the ball touches, changing the training regimen, use of coach encouragement, changing the type of marking, using floaters, and other artificial rules.”

  1. 70. Avoid using “among” twice. 

Reply:

We replaced the first use of the word with ”from”.

L.77. Please consider changing “modify” change.

Reply:

We used ”change” as a substitution for modify, as suggested.

L.98. Add a full stop after the reference. 

Reply:

We corrected and added the full stop.

While the introduction is well-written with relevant literature, I am missing more information regarding youth players. I mean, the introduction is solid but apart from that SSG helps youth players to develop technical skills, it is not clear why will the authors study this age group. 

Reply:

We added a new sentence to further explain the reason to study youth soccer players: ”According to Snow, youth players have an intrinsic desire to play the game and activities should be game-specific because they are fun and age appropriate”.

Methods. 

L140. Please provide additional information regarding the participants. What was their height, weight, and playing experience? 

Reply:

We added the details about height, weight and body mass index.

L.154. What initial tests? Is there any difference between the players’ profiles between groups?

Reply:

We added the phrase based on the ”aerobic capacity initial testing results”.

We used univariate ANOVAs to analyze pretest differences between experimental groups. The results showed that there were no differences between the groups at baseline in terms of aerobic capacity (F = .11, p = .745), total distance covered (F = .194, p = .666), total sprint distance (F = .016, p = .900, task-oriented motivational climate (F = .008, p = .931), or ego-oriented motivational climate (F = .738, p = .405). Our analysis suggest that our experimental groups were equivalent before the intervention.

  1. 156. Please provide a reference for the validity of this system. 

Reply:

We added a reference about the validity of GPS devices and technical specifications of the model.

L180-182. Could it be expected that players improved their performance, even without the intervention, as it is likely that being exposed to training routines (as those since the start of preseason) would contribute to physical fitness enhancement?

Reply:

We started applying the intervention at the beginning of the preseason, when the athletes had a low level of physical training. Athletes were not exposed to other aerobic capacity training apart from SSGs. However, we understand that the other workouts may also have contributed slightly to the development of exercise capacity. Using the repeated measures design, we attempted to control for these aspects.

L.190. Please consider including a table with the games performed, or a figure describing the protocol procedures. 

Reply:

We added Table 1 to summarize the protocol procedures.

  1. 195-202. Which type of rules were used in the SSG? Was there any specific information from the coach? Did it contain goalkeeper? Further information on how these SSG were designed is required. 

Reply:

We added the following paragraph:

”The games were played at small goals, with no goalkeepers. No other task constraints or rule modifications were used. The coach explained to the players that the aim of the games was to develop effort capacity and emphasized the importance of effort exertion during the games.”

L230-246. My main concern here, although I do understand its relation to the authors' study aim, is that running more it is not always running better from the SSG perspective. For reference, please see: 10.1080/15438627.2017.1393754

Reply:

Although tactical training plays an important role in practicing the game of soccer, our study was not aimed at this aspect.

There is the possibility that participation in SSGs that emphasize the development of effort capacity will negatively affect tactical aspects such as field positioning. However, given that our games consisted of few participants (2 or 4), we did not consider this to be a concern. In the after mentioned study, Folgado et al. used 8 vs. 8 +goalkeepers large-sided games in which tactical performance becomes more relevant.

L.240. What is considered a high level of effort? What was the level of the coaches? Were they able to perceive the appropriate time to encourage the players? How was this controlled by the research team?

Reply:

We substituted ”a high” with an ”increased level of effort, e.g., a sprint or a high intensity duel”.

The observations about effort expenditure were subjective because they were dependent of coach judgement. However, the coaches knew the group of players well and had about 7 years experience in coaching, therefore we believe that they were able to properly appreciate when one of the players put in a high effort. In order to control for the coach encouragement, they were instructed before by the research team and the main researcher was present in every training.

Results 

Please consider including the percentage variation change between each pre-post measurement. 

Reply:

We added the percentage variation change.

Discussion

L-328-329. Did the authors control the time performed on friendly and competitive matches by the players from both groups? Were there any differences? Or even maturational differences?

Reply:

We did not control for time performed on friendly and competitive matches.

However, we believe that the randomization contributed to obtain two homogeneous groups in terms of time performed also. On the other hand, we understand that the small number of participants could affect the results, which was presented as a limit of the study.

Reviewer 3 Report

Dear authors,

Congratulations

Thank you for the research you have done on SSG, which is one of the important training components of today. I have studied your research in detail and found your article interesting. The only question in my mind about your research was the number of subjects, which you gave as a limitation at the end of your research. However, if possible, the number of subjects in normal conditions would provide clearer information for the reliability of the results, I would appreciate it if you add it to your research. Finally, there are some minor english grammatical errors, I'd appreciate it if you could check. Your research is currently eligible for publication in the ijerph journal. 

Your sincerely

Author Response

Comment:

Congratulations

Thank you for the research you have done on SSG, which is one of the important training components of today. I have studied your research in detail and found your article interesting. The only question in my mind about your research was the number of subjects, which you gave as a limitation at the end of your research. However, if possible, the number of subjects in normal conditions would provide clearer information for the reliability of the results, I would appreciate it if you add it to your research. Finally, there are some minor English grammatical errors, I'd appreciate it if you could check. Your research is currently eligible for publication in the IJERPH journal.

Reply:

We added the sentence ”Brysbaert  recommends a number of 27 participants in each group to achieve high power”. We searched for grammatical errors and made correction if that was the case.
